# How the COVID-19 Pandemic Will Change Workplaces, Healthcare Markets and Healthy Living: An Overview and Assessment

Heather Kolakowski, Mardelle McCuskey Shepley, Ellie Valenzuela-Mendoza and Nicolas R. Ziebarth *

Cornell Institute for Healthy Futures, Cornell University, Ithaca, NY 14853, USA; haf3@cornell.edu (H.K.); mshepley@cornell.edu (M.M.S.); ev227@cornell.edu (E.V.-M.)
* Correspondence: nrz2@cornell.edu

**Abstract:** The COVID-19 pandemic has disrupted most aspects of our lives: how we work, how we socialize, how we provide health care, and how we take care of our most vulnerable members of society. In this perspectives article, we provide a multidisciplinary overview of existing research covering these fields. Moreover, we enrich this research overview with news reporting and insights from a panel of expert practitioners affiliated with the Cornell Institute for Healthy Futures. We sketch existing evidence, focusing on how the pandemic has transformed our lives since March 2020. Then, for each of the fields covered by this article, we propose optimistic perspectives on what healthy living could look like in the future, given the current challenges and opportunities. In particular, we discuss the needed transformations of our workplaces, the health care market, senior living, healthy eating, and personal wellness.

**Keywords:** COVID-19; healthy futures; future of labor; hospital markets; senior living; infections; spread of diseases; social justice; racial inequality

## 1. Introduction

The COVID-19 pandemic has disrupted and affected most aspects of our lives: how we work, how we socialize, how we provide health care, how we take care of our most vulnerable and disenfranchised members of society, and even our rituals of death and mourning. This paper considers the post-COVID-19 era, focusing on how we as society could design healthy futures.

How do we work towards an equitable society with universal suffrage? This is the question facing individuals, communities, and organizations throughout the United States and worldwide. Consolidating a plethora of scholars and experts, we begin by outlining the growing trends in telework, state and federal legislation, the rising focus on improving working conditions, and the common goals across industries for healthy employee outcomes. The innovations have already begun, changing our perspectives on mental as well as physical well-being, and the very notion of "work" itself. This section likewise emphasizes the impact of social unrest as a leading factor for on-going discussions in the workplace regarding racial justice, a living wage, and equity in relation to job prospects.

The next sections continue to summarize the multidisciplinary literature on how COVID-19 has transformed the health care sector and senior living markets. To provide a broad in-depth analysis, we not only refer to classical peer-reviewed research, but consider opinions of expert practitioners as well as regular reporting by newspapers and magazines on case studies. Considering all this evidence, the authors make predictions and suggestions about how this pandemic could lead to healthier living in the post-pandemic era.

We continue the conversation with a consideration of how food insecurity impacts health outcomes. This section emphasizes the challenges in addressing food shortages in

schools and broader communities in a post-COVID-19 world in the United States. We offer suggestions on how to meet these challenges, connecting ultimate success of eradicating food insecurity with social justice initiatives for equitable employment opportunities and a living wage.

Overall, this article contributes to the fast-growing literature on COVID-19 in several ways. First, it is an attempt to synthetize our current knowledge across several fields in the social sciences, in line with the mission of the Cornell Institute for Healthy Futures (CIHF), the first academic center in the United States to combine hospitality, environmental design, and health policy and management. Second, the article not only incorporates peer-reviewed published research but also explicitly relies on news reporting, anecdotes, and a summary of expert opinions of select practitioners across the fields of health care, hospitality, senior living, and labor. This combination of published research, reporting and qualitative insights allows us to cast a wide net of information leading us to several predictions and outlooks of how the post-COVID world might unfold. Throughout the manuscript, we try to engage in a positive future and life-affirming discussions that do not shy away from societal fractures such as racial inequality. Hopefully, it will inspire further connections and innovative approaches to produce healthier futures for all of us.

## 2. Healthy Work in the Post-COVID-19 World

The novel coronavirus pandemic has disrupted labor markets and the way we work. At the beginning of the pandemic, in February and March of 2020, a massive stock market decline coincided with government lockdowns and stay-at-home orders. Millions of American workers were laid-off or furloughed. By April of 2020, the unemployment rate had quadrupled to 14.7% and weekly applications for unemployment benefits were 10 times higher—at almost 7 million—than during the highest week on record to that date in 1982 [1–3]. As a consequence, the CARES Act injected 2 trillion dollars into the economy to prevent a collapse. The bill consisted of USD 1200 checks sent to households with annual incomes of less than USD 99 thousand, funds to small businesses, and payments to state and local governments [4,5]. In addition, the bipartisan Families First Coronavirus Response Act (FFCRA) was signed into law on 18 March 2020. FFCRA included, for the first time ever, a federal mandate for enable employees to take up to two weeks of paid sick or family leave due to COVID-19 [6,7]. Additional bills and rescue packages followed.

At the beginning of 2021, the pandemic raged worse than ever before. Every day, the United States counted more than 200 thousand reported infections and 2000 new COVID-19 deaths [8,9]. Since that peak, cases have dramatically declined because of swift policy action at various levels, such as the rollout of the biggest vaccination program in U.S. history—the United States administered 200 million vaccine shots during the Biden administrations' first 100 days in office. The stock market has regained new highs, millions of workers returned to work, and the unemployment rate has decreased to 5.4% in July of 2021—but this is still significantly above the 3.5% rate in February of 2020 [1,2].

Many of us expect(ed) a return to "normal" by the fall of 2021. It will be a "new normal". This is particularly true for the way the pandemic has altered how we are working and living. It will also alter how we work and live in the future. Discussions within the CIHF community of experts produced forecasts for the ways in which COVID-19 will change the culture of work in the United States and around the world.

### 2.1. The Lessons of How Contagious Diseases Spread Will Change Our Work

Remember the days when it was fine—in fact, a sign of a superior work ethic—when your co-workers came to the office sneezing and with runny noses? Not just in the U.S. but around the world, workplace cultures were built on the false premise that working sick and being present would demonstrate determination, resilience, and an exemplary approach to productivity. Social norms and peer pressure ensured this approach became engrained within work culture [10]. This has changed thanks to the pandemic. Legislatures will pass paid sick leave bills and companies will revise their attendance policies. Customers will no

longer tolerate sick employees serving food and will shun businesses that fail to enforce safety protocols. We debate how to overcome vaccine hesitancy and whether employers should have the right to force their employees to get vaccinated, especially in the health care sector. In the post-COVID world, it will be common to wear masks, practiced in Asia for years, especially during the flu season [11]. You will see more disinfectants when entering stores or your workplace and people may be reluctant to shake hands.

The new workplace normal and infection awareness will also have tremendous implications for future office design. The silver lining is that workplaces will become safer and better prepared for another pandemic as well as flu seasons. The new normal will involve frequent disinfecting of commonly touched surfaces, hands-free doors, socially distanced spaces, and touchless transactions. It will reduce the number of sick people at work and reduce flu-related deaths [12,13]. The lessons learned will lead to healthier workplaces as well as healthier employees and customers.

### 2.2. The Pandemic Will Be a Game Changer for Telework and Online Conferences

We will travel again and meet in person, go to bars and restaurants, and have in-person conferences. However, telework, online conferences, and meetings will continue to boom [14,15]. The pandemic has been a game changer for online meetings. Why spend millions of dollars on travel budgets, valuable work time and energy, if we can hold 2-hour meetings via Zoom? There is simply no reason to continue the wasteful and environmentally damaging excessive travel that was completely acceptable prior to COVID-19 [16]. The pandemic has shown us that meetings can be quick and simple. Certainly, technical glitches will remain and Zoom fatigue is real. Yet, in the future, we will become more sophisticated and pragmatic when weighing the benefits and costs of work travel, converging on a hybrid of in-person and online meetings. As degradation in demand will very likely be concentrated in the corporate segment, it will have profound implications for future hotel and resort meetings and their design as discussed below.

Likewise, telework will increase. Employers' attitudes will continue to shift and employees will increasingly expect and demand flexible work arrangements [17,18]. Some companies have already gone so far as to eliminate their in-person offices and move entirely to telework. They realized how much money they could save divesting themselves of expensive office space in downtown locations [19–21]. Instead, why not encourage employees to work from home and let them set up home offices? Some employees may prefer that model. Others will not.

In the spring of 2020, in the United States it became increasingly clear that due to a lack of childcare options, many employees have not returned to their previous jobs and left the labor market, at least temporarily [22]. The lack of paid leave coverage has been cited as a main reason for the difficulties of U.S. employers to fill jobs; time use data from the Bureau of Labor Statistics show that the share of employees who juggle work from home and child care has increased tenfold during the pandemic [23]. Companion evidence shows that unmet sick leave needs tripled during the pandemic compared to previous years [24].

### 2.3. Our Appreciation of Essential Workers Will Remain and Improve Their Work Conditions

According to the U.S. Department of Homeland Security, essential workers "are those who conduct a range of operations and services that are typically essential to continue critical infrastructure operations" [25,26]. During the pandemic, the broader population learned that essential workers are critical to society's overall well-being. Broadly speaking, essential workers include employees in the sectors of child care, long term care, agriculture and food production, transportation, "critical retail" such as grocery store workers or mechanics, as well as "critical trades" like construction workers or plumbers [25]. The pandemic taught us how we depend on essential workers: our cashiers, nurses, teachers, and public servants.

Numerous newspaper stories and documentaries have discussed the working conditions and everyday tasks of essential workers [27]. We have applauded and demanded

better working conditions for them. Essential workers felt appreciated, some for the first time [28,29]. In addition to a confidence boost, it made them realize that their work is *essential* for society to function. Hopefully, this appreciation and acknowledgement will become permanent and help attract young and talented people for much demanded professions. Medical school applications have increased, but one-third of health care workers consider leaving the profession [30,31].

Still, there is hope that essential workers will not just receive applause, but also higher salaries and better fringe benefits—either through market-driven changes or policies raising minimum wages and passing bills that guarantee paid sick and family leave. As women, minorities, and disadvantaged populations are overrepresented among essential workers, better working conditions would also reduce racial and economic inequalities [32,33].

Already today, we begin to see first signs that employees may have stronger bargaining power and will demand better fringe benefit and flexibility, as many employers seem to struggle to hire employees for low-wage jobs, especially in the hospitality and accommodation industry [34]. If we continue to see labor shortages, one economic implication is a market-driven increase in wages and better working conditions.

### 2.4. Policymakers Will Pass Legislation to Improve Work Conditions

With the new Biden administration, the next four years will bring several legislative initiatives to improve working conditions in the United States. On the Democrats' agenda is a higher minimum wage, better workplace protection, a transformation of the energy sector, higher subsidies and easier access to health insurance, paid sick and family leave, and better protection for LGBT+ communities [35–38]. The recent expansion of the Child Tax Credit is a serious bolster for working families in need of childcare support. Obviously, the question of whether bipartisan initiatives will be successful as well as the outcome of the 2022 mid-term election, will determine the success of the White House agenda. On the other hand, following the U.S. tradition, lower legislative levels will pass plenty of bills and change will not necessarily depend on a united Federal response.

### 2.5. Awareness for Racial and Economic Inequality Will Lead to Slow but Steady Improvements

In the midst of the pandemic, the murder of George Floyd was a stark reminder that racism is inherent in U.S. society. In fact, the pandemic itself highlights entrenched racial inequality; the pandemic has hit communities of color hardest and increased economic, social, and racial inequities [39–42]. Inequality in all its forms continues to be shared widely in news and social media. Due, in part, to the high visibility of the Black Lives Matter (BLM) movement, corporations have begun to address social justice and racial equity through Diversity, Equity and Inclusion (DEI) initiatives. Increasing diversity training have become—for now—part of corporate culture; in fact, numerous companies seemingly embraced a sense of responsibility in combatting racism itself [43]. In addition, U.S. states like Washington and California passed mandates for gender-diverse company boards [44].

In the United States, many hope this is a true turning point. The combination of historic failures in addition to the social shock that resulted from circulated videos of police brutality towards people of color have shifted individual and organizational viewpoints. While voter perceptions of BLM were critical until 2018 and net voter support was in fact negative, the movement's popularity has grown steadily since and shot up to an all-time high after the inception of the pandemic and the murder of George Floyd [45].

## 3. Healthy Lifestyles and Wellness in the Post-COVID World

In addition to its impact on the way we work, the novel coronavirus pandemic will also have a lasting impact on our lifestyles more generally. Some of the predictions we make for the labor market naturally carry over to our private lives:

### 3.1. The Lessons of How Contagious Diseases Spread Will Increase Our Health Consciousness

Similar to how our increased knowledge about the spread of contagious diseases will affect the organization of our work, it will have a profound impact on our social lives. We will become less tolerant to socialize with sick people. We will shake hands less often. We will wear masks more often voluntarily and use more disinfectants, during and beyond flu season. Technology-driven home entertainment will also drive how and where we spend our leisure time.

Moreover, the pandemic taught us to take health risk factors such as high blood pressure, diabetes, and being overweight more seriously [46]. Wearables—fitness trackers—became prolific within the last decade long before COVID-19. However, the shock of 2020 will certainly reinforce and speed up existing trends. Valuing fitness and leading a healthy lifestyle are such trends. Real-time tracking of our individual and environmental health risks will become more prevalent than ever before, while other socio-demographic groups will continue to lead unhealthy lifestyles and reinforce the existing health gradient [47].

### 3.2. Our Awareness of the Importance of Wellness and Mental Health Will Increase

Likewise, when we think back and remember the dark and lonely days in isolation, we have learned to appreciate social relationships and in-person get-togethers again. The value of time with our families and friends is more salient to us. There is a reasonable chance that our hectic pre-pandemic lives might even improve post-pandemic. Maybe we understand that "less is sometimes more" and learn to declutter our overburdened schedules. There is a chance that we will spend our time more wisely, and focus on the people and activities that are really important and meaningful to us. Organizational changes to our work, such as more telework and less travel, will certainly help to accomplish an increased focus on wellness and mental health.

However, while such an optimistic outlook will apply to some, not all of us will be better off in the post-pandemic world. We know that unemployment leaves long-lasting mental and physical health scars, not just for those who are laid off but also spouses and former co-workers [48,49]. The pandemic dramatically increased domestic violence [50], put severe mental health burdens on many of us [51], and will likely increase separations and divorces [52,53]. These consequences appear even more dire when we consider the millions of deaths [8], "Long COVID"—the possible long-term health effects on survivors [54], and the cumulated effects on their relatives and friends. Nobody can predict how long it will take to overcome such tragedies. What we can hope for is an enduring legacy of critical lessons learned about our physical and mental wellbeing throughout this challenging time.

### 3.3. We Will Continue to Travel, but Less Often and Less Far

The pandemic forced almost everyone to travel less, and less often. Many of those cancellations were undesired and painful, but others were not. In fact, the pure necessity to stay local and explore our own state and communities may have brought some awakening to us. Happiness can be right in front of you. Some have made new social connections and friendships that will last longer than the pandemic.

In addition, safety considerations and travel restrictions will become a more decisive factor and shape pricing and the industry's profitability for years to come. For example, more than a year into the pandemic, countries around the world have still imposed heavy travel restrictions for tourists—such as the United States as well as the European Union but also countries like Australia and Canada.

In the U.S., we have already witnessed people leaving their expensive downtown apartments and moving to rural suburbs, which are cheaper, quieter, and less crowded [55]. Observers have called out the "Death of Cities" [56]. Trends toward more health consciousness and more mindfulness will likely affect how and where we live and how much time we spend in cars and on planes. It is almost certainly less, not more, time.

## 4. Healthcare in the Post-COVID World

COVID-19 affected all segments of the health care industry; some effects will be temporary, but others are likely to have profound long-term consequences:

### 4.1. The Hospital Market Concentration Will Further Increase

In 2020, health care demand plummeted by about a quarter [57]. This was the case for "elective" treatments but, surprisingly, heart attack and stroke admissions also plummeted [58]. People were afraid of seeing their doctors or going to the hospitals. At the same time, COVID-19 hospitalizations skyrocketed and a lack of health care personnel forced hospitals to re-organize themselves [59,60]. The government put up emergency funding to provide liquidity and to avoid the collapse of an entire sector of the economy [61].

Still, it is very likely that many hospitals will not survive the pandemic and will have to close or merge with bigger entities [62–64]. The existing trend toward more market concentration will further accelerate, and access to inpatient care further deteriorate, especially in rural communities.

### 4.2. Telemedicine Has Seen Its Break-Through

On the bright side, the pandemic likely brought about a break-through for telemedicine. Estimates suggest that a quarter of all treatments accomplished via video chats, especially mental health treatments and first assessments by primary care physicians [65,66]. Reimbursement coverage for telemedicine by public and private insurers will likely remain in place. In the meantime, patients and providers will have become accustomed to the convenience of new technological opportunities [67]. Patients and providers will increasingly use these new telemedicine options; this new approach will benefit highly educated, tech savvy young professionals as well as rural populations without sufficient access to health care facilities.

### 4.3. The Vaccine Development Will Be a Lasting Success for Scientists and Regulators

Never before in human history have we as societies devoted so much money, time, and energy to the development of a single vaccine in such a short time [68]. Still, the breathtaking and record-breaking speed with which scientists have developed several highly effective vaccines is a major success story [69,70]. Although a percentage of the population mistrusts the vaccine rollout, nonetheless, vaccines are already saving millions of lives and avoiding hundreds of millions of infections; vaccines are also inspiring generations of young women and men, reinforcing our trust in science, and boosting the confidence of what humans can accomplish in a well-regulated and incentivized health care market.

### 4.4. Contagious Disease Safety Protocols Will Improve Our Quality of Care

A final optimistic outlook is based on the presumption that providers and health care organizations have drawn lessons to be prepared for the next wave or pandemic—or simply the next heavy flu season. Perhaps nothing has been more shocking than the fact that nursing home residents over-proportionally died of COVID-19 [71], in many cases alone and without family and friends. This has been true all over the world. Some countries and regulators learned faster than others but, in general, a major focus in this pandemic has been on how to protect the most vulnerable of our society while providing them the dignity and human interaction they deserve and need [72].

During the pandemic, testing was rolled out with a focus on nursing homes and inpatient clinics, and contagious disease protocols were revised and revamped [73]. Nursing homes, the elderly, and health care workers have been prioritized in the mass vaccination campaigns around the world [74]. All these efforts did not entirely prevent deadly outbreaks and, often, policy and management action were inadequate and led to preventable suffering, deaths, and tragedies [75,76]. In the post-COVID world, however, the hope is that we will be better prepared for the next pandemic with improved safety standards to reduce the spread of common contagious diseases.

## 5. Senior Living in the Post-COVID World

The pandemic impacted all aspects of the healthcare continuum, including the senior living sector. During the pandemic, innovations and modifications were needed to manage the challenges of congregate care settings as well as the social wellbeing of both residents and care givers. Some of the policies and procedures will be lifted as vaccines are more fully accepted and administered; however, even as we move forward, the planning for the possibility of a future pandemic needs to be considered. Nonetheless, there are other innovations that will remain in the future as senior living enters into the new normal. For the purposes of this section, the term "senior living" is specifically focusing on the portion of the industry defined as "where the needs-based private pay senior-housing industry provides assistance with activities of daily living and socialization in a residential setting" [77].

### 5.1. Growth Will Continue into the Future

The "senior" segment of the population will grow more than three times faster than over the past decade. "Between 2020 and 2025, the 75 and up population will increase by 23% or by 1 million-plus annually" [77]. In addition, the changing demographic of seniors regarding chronic conditions such as Alzheimer's, obesity, and diabetes will require more personal care treatments. While the growth in this population will mean an increase in demand for senior housing, it also offers the opportunity of an added value proposition of bringing care into the home. The concept of "aging in place" has also been highlighted during the pandemic, when seniors were more fearful of entering into communal living situations due to the spread of COVID-19.

One significant challenge is the lack of potential caregivers among the younger population. "U.S. Census Bureau projections forecast the ratio between 18–64-year-old and 85 and up populations to decline by 9% between 2020–2025" [77]. The ratio of caregivers (45–64-year-olds) to those over 80 will shrink from 7:1 to 4:1 by 2030 [78]. This presents a current and future labor shortage within the industry.

Argentum—one of the largest associations in the assisted living field—reported that by 2025 there will be a need for 35 thousand positions in management and business/financial operations and nearly 1.2 million jobs overall [79]. A symposium of leaders from a variety of organizations (including universities, associations and major providers) titled "Vision 2025" concurred that there was a major manpower requirement that would need to be addressed. The impact of COVID-19 has further highlighted these challenges.

### 5.2. The Potential for Internet-of-Things (IoT) Monitoring Solutions

While demographic change and the industry structure will likely exacerbate the labor shortage among healthcare workers in the nursing home and senior living market further, the use of technology will certainly gain in relevance. Even before the pandemic, industry pioneers pointed out the potential for an increased use of monitoring applications and smart devices, both in nursing facilities and the outpatient senior living market [80–82]. In the future, such technology will not just provide supplementary assistance to facilitate recruitment and training of healthcare workers, but actually support clinical monitoring and decision making. As discussed in Section 2.2, COVID-19 brought the breakthrough in telemedicine. Consequently, along with smart monitoring, it will likely be a game changer for the senior living market.

However, while the use of such technology offers great potential to improve clinical outcomes and the quality of life of people who need assistance, concerns about privacy, legal barriers as well as barriers to digital use among the older generation may increase the time to full adoption of smart technologies [83,84]. In addition, cultural barriers will likely result in unequal diffusion of robot use in elderly assistance over time [85,86].

### 5.3. Socialization Will Continue to Be Critical to Satisfaction

Socialization is a primary driver for seniors who look to congregate care settings. Even for seniors who want to age in place, socialization and social interactions are critical for health and well-being. Significant research has been conducted on the importance of human connection for quality of life, even before the impacts of the pandemic and social distancing measures. Social isolation and loneliness can increase mortality rates among seniors, so the "health protective benefits of social distancing must be balanced by the essential need for sustaining social relationships" [87].

With this factor in mind, many senior living facilities have focused on ways to maintain social connections amongst their residents while maintaining safety protocols during the pandemic. For example, at Brandywine Living at Dresher Estates, Rachel Kaufman reinvented their activity program: from internal interactions between residents and staff, such as "exercises, bingo and flower arranging in the halls" to using Zoom for horticulture, exercise classes, bingo, and religious services [88]. Other facilities created outdoor spaces for families to visit with relatives, using patios and other seating areas to allow for greater social distancing. Another approach by CIHF's Industry Scholar, Meredith Oppenheim, has been the creation of virtual communities and activities that can be done from the individuals' room/apartment/home via the Vitality Society™.

### 5.4. A Greater Focus on Healthcare in Residential Senior Living Models

There has been a debate between the importance of healthcare versus hospitality in senior living. Historically, there has been a focus on customer service and hospitality aspects—especially in the independent living segment and in less care-intensive assisted living facilities—rather than healthcare. Since the passage of the Affordable Care Act (ACA), this focus has shifted to an improved coordination between the acute and post-acute providers. Efforts to revamp coordination were also driven by penalties for readmissions. Some providers, such as Juniper, have worked to implement programs to improve coordination, and researched and published the benefits of programs such as their "Connect 4 Life" model.

COVID-19 has helped further shift the value proposition. Effective safety and medical care programs and coordination has become more important during the pandemic [89]. Hospitality-like services and customer service continue to be valued, but post-COVID, they are no longer enough. "Providers must accept that their communities are home for a population that has significant care needs and is especially vulnerable to infectious diseases" [89]. In addition, a shifting mindset from reactionary to preventive care can help "avoid hospitalization and stop a downward spiral for a resident" [90]. Utilizing predictive technologies can assist with fall reduction, infection control, hospital visit reduction, mental health care, employee accountability and occupancy rates [90].

Families and residents now realize that care can be delivered where residents live. However, during a recent interview for CIHF and eCornell, Bob Kramer—founder of the National Investment Center for Seniors Housing and Care (NIC) and head of Nexxus Insights Consulting—noted that it also requires a strong team-based culture with a true valuing of the front-line caregivers—frequently referred to as the "heroes" of the pandemic. This applies whether in senior living communities or home-care organizations providing in-home services. All types of seniors housing and care services will face labor challenges, as noted previously, so building the kinds of cultures that will attract and retain excellent caregivers and other staff is critical to providing high quality care services.

### 5.5. Changes Will Occur in Federal and Other Oversight and Regulation

Regulation in the skilled nursing facility (SNF) sector, also known as nursing homes, has standards set by the federal government. While Continuing Care Retirement Communities (CCRCs)/Life Care Communities are designed with a full range of services from independent through SNF, other senior living providers that do not offer skilled nursing services have traditionally had regulations that vary significantly across states. The

current Federal oversight to the assisted living industry is primarily focused on Medicaid-certifications, as well as federal labor and occupational safety laws [91]. With the impact of COVID-19, there will be more moves to standardize requirements to meet /needs in different environments. In Senior Living News, Mullaney (2021) indicated that while there are costs to comply with additional regulations and other impacts, this might have some positives: "Regulation could even bring some benefits in terms of standardizing the industry and clarifying to consumers what assisted living means" [89].

At a National Investment Center for Seniors Housing & Care (NIC) fall 2020 conference, Anne Tumlinson, founder and CEO of ATI Advisory and Avalere Health Founder, and former CEO Dan Mendelson, both agreed that the senior living industry needs to mobilize and be proactive in shaping the regulatory framework at the federal level. According to Tumlinson, "the industry must be ready with what it thinks is in the best interests of the residents and the overall health of the industry" [92].

Given the mistakes made throughout the pandemic, regulators, and consumers will demand increased transparency regarding policies and health standards. During the pandemic, when many senior living facilities implemented a no-visitation policy to stop the spread of COVID-19, communication became even more critical. Face to face interactions decreased and communication channels suffered. Creating a system to connect with families in day-to-day messaging requires time, organization, and training, which was in short supply during the pandemic.

With strict HIPAA laws, any medical information about residents has to be carefully communicated only with people that are directly involved with the care of the individual. Many organizations have worked out at least interim solutions, using FaceTime^TM and other applications with tablets and other technology. Others have already adopted more sophisticated solutions, by creating regulatory compliant, integrated, and reliable systems to clearly communicate with patient families; in the future, this must be a priority for senior living communities, independent of the potential of another pandemic.

*5.6. Competition Will Increase and Different Residential Models Will Be Pursued*

The challenges associated with operations among social distancing restrictions demonstrates that only strong operational companies will thrive in the future. "(O)lder adults have more options for where they live and receive care, such as: co-housing communities; with family members who are now working from home full time; and in their own homes, supported by the ever-widening array of on-demand services that they became adept at using during the pandemic" [89].

Even in Ithaca (NY), we have seen ideas like "Love Living at Home" and other networks to support aging adults who want to remain at home. Additional models include the Naturally Occurring Retirement Communities (NORCs) or the village concept. An early example is the Beacon Hill area in Boston where neighbors help with chores and grocery shopping allowing seniors to remain in place longer. Especially with COVID-19, many people have delayed the decision to move into senior housing to avoid institutional settings. People are likely to continue to investigate alternatives. Home care is an important part of that, although the costs are starting to go up dramatically.

Boomers who were part of the COOP movement in the late 1960s to early 1970s will also be attracted to the sense of community that these models provide, as long as they continue to have their mobility and cognitive function. Bob Kramer noted that some of these trends include apps that connect people with common interests to pool their resources for co-housing. For seniors who maintain reasonable mobility and cognition, these options for community-building allow them to relive, in a sense, their college years where communal living provided socialization and support.

Another possible development could be moving to pod or small house models lessen the burden of isolation or serious infectious disease outbreaks. A recent article on small house communities for low to moderate income residents was cited as a possible framework

to adopt for seniors housing as noted in the post "Tiny House Villages—A Viable (and Affordable) Option for Senior Living?" [93].

The Program for All-Inclusive Care for the Elderly (PACE) provides an interesting model. It functions like a small accountable care organization where the PACE program manages a population of seniors and is paid a set amount per month to oversee their clients' care needs. A multi-disciplinary team manages the cases and works to allow residents to stay in their homes and avoid hospitalizations and nursing home admissions. They also use a medical-model adult day center as a pivot point where residents can be transported for services and meals if needed. This program is generally focused on SNF-eligible Medicaid and Medicare beneficiaries, many with both or "dual-eligibles" managing their cases to avoid high-cost environments. It is unclear whether this will be expanded by Medicare and Medicaid, although historically, a small program run by local nonprofits is possible since at least one organization has just successfully executed a USD 350 million IPO—InnovAge (Nasdaq:INNV) indicating wider availability for the future.

### 5.7. Technology Will Be a Source of Innovation

Many industries increased their technology applications during the pandemic to adjust to contactless interactions with their consumers. Maintaining safety as well as connectivity became the balancing act requiring investment in hardware and software across many sectors. While it might seem like a base starting point, senior living "communities clearly must have robust infrastructure to support stable connectivity in all locations" in order to support increased usage of digital Wi-Fi networks [89]. This will include more delivery of care applications including remote monitoring and telehealth. Using wearable devices for residents and caretakers to help with contract tracing and infection control is one aspect of data intelligence with multiple applications [94]. We will also see new applications of technology for engagement opportunities [95].

As technology innovations to provide enhanced level of care for residents expand throughout the senior living sector, the pandemic also demonstrated an increased use of digital offerings that many operators relied upon. From a marketing and sales perspective, utilizing virtual tours and open houses to show case the facility to potential residents and their families became the primary method of information sharing when facilities closed to on-site guests. While resident intake might have slowed during the pandemic, there is significant pent-up demand for senior living facilities that will increase intake numbers as vaccine distribution continues.

Beyond digital marketing, the digital innovations will be about balancing safety, efficiency and vitality. Some applications are providing opportunities for programs that make a positive impact on quality of life, including options to connect with family members that are long-distance, or when a resident might be too sick to travel.

## 6. Food Access in the Post-COVID-19 World

Before the pandemic began, more than 35 million people, including nearly 11 million children, lived in a food-insecure household in the United States [96]. According to Feeding America, the number of people who are food insecure could rise to more than 50 million, including 17 million children [97]. Even more distressing is the potential long-term effects of the pandemic on food access in the U.S. "It took ten years for food insecurity rates to return to pre-Great Recession levels. For now, with no immediate end to the crisis in sight, demand for charitable food assistance is expected to remain at elevated levels for the foreseeable future" [97].

The American Rescue Plan Act of 2021 (ARP) was passed on 6 March 2021. This USD 1.9 trillion COVID-19 relief bill helped bolster nutrition assistance across the country by extending the 15% boost to Supplemental Nutritional Assistance Program (SNAP) benefits through 30 September 2021, and provides additional administrative funds to extend the SNAP benefits as well as expand access to SNAP online purchasing.

In addition to this support, the bill also extends the Pandemic Electronic Benefit Transfer (P-EBT) program; this extension provides grocery benefits to replace meals that children miss when schools are closed. It also helps school-aged and young children through the summer and 2022 academic year to access healthy meals, as well as support nutrition programs for older adults and Native American communities under the Older Americans Act [98].

Other critical support of the bill includes investing USD 3 billion to help women, infants and children access food through increased outreach for the Special Supplemental Nutrition Program for Women, Infants, and Children (WIC), and provides U.S. Territories with USD 1 billion in additional nutrition assistance. Finally, the ARP also provides partnership funds for restaurants to help feed people and to provide jobs for laid-off restaurant workers [99].

The economic and social impact of the pandemic can be felt in many ways across various aspects of the food system, intensifying existing socioeconomic disparities. The following projections focus on food access and the food system in the post-COVID United States.

### 6.1. Demand Will Remain High for Food Banks and Charitable Services

During the pandemic, people who had never experienced food insecurity before began relying on food banks. Between March and June 2020, 4 in 10 people visiting food banks were there for the first time [100]. "In October (2020) alone, food banks distributed 50 percent more food than they did last year at the same time" [100]. In addition, Black, Latinx, and Native American households struggle with inequities in housing, employment, and education [101]; food insecurity disproportionately impacts these communities [96]. Access to food through the food banking system of local food pantries is often easier and faster than applying for SNAP benefits or other government aid. Food banks have increased their capacity throughout the pandemic; many expanded their delivery and marketing methods through drive-thru pantries and social media announcements.

However, there are still challenges with supply chain delays, volunteer shortages due to social distancing guidelines, and the economic impact of the increase in food prices. Once the pandemic subsides, and businesses and local economies open up, the demand for charitable food assistance will slowly decline. However, there could still be an elevated level of need for many years to come, as evidenced by the 10-year lag in food insecurity rates following the Great Recession in 2008 [102]. While charitable groups and food banks have increased their capacity to address the increased demand for food, they cannot fix the problem by themselves. For every meal that the food banks provide, SNAP provides nine meals [103] therefore, continued federal and state funded programs will be critical in the future.

### 6.2. Need and Support for SNAP Benefits Will Increase

Throughout the pandemic, various legislation helped to waive certain aspects of applying for SNAP benefits, as the program saw increased participation due to rising unemployment as well as longer processing times resulting from social distancing measures within the service departments. According to The Center on Budget and Policy Priorities, "(a)vailable data suggest that 6–7 million more people have applied and been approved for benefits since February (2020), a 17 percent increase nationally" [104]. With each round of approved legislation for COVID-19 relief, the nutritional assistance programs received support. For example, most recently, for the January–June 2021 period, the maximum benefit allowed was increased to 115 percent of the June 2020 value of the Thrifty Food Plan, through the Consolidated Appropriations Act, 2021 [105].

However, once the temporary relief measures expire in 2021, it will be the responsibility of Congress and the USDA to continue to support the program [106]. The temporary measures put in place to help the low-income family's access the SNAP benefits in a timelier manner will expire, causing challenges for those families still trying to recover from economic hardships created by the pandemic.

The recent support to SNAP through ARP signifies the focus of the Biden administration to address the growing hunger crisis in America. Agriculture Secretary Tom Vilsack even noted in his keynote address at the National Anti-Hunger Policy Conference in March 2021 the importance of support and expansion of these programs, including increased messaging to help people understand what these programs are designed to do [107].

*6.3. Expansion of the EBT Acceptance for Online Delivery of Groceries Will Occur*

The 2014 Farm Bill mandate established policies that included domestic food assistance; as a result, a pilot program launched in 2016 to test the feasibility and implication of allowing retail food stores to accept SNAP benefits through online transactions. "For households to make online purchases, the online shopping and payment pilot is required to be secure, private, easy to use, and provide similar support to that found for SNAP transactions in a retail store. Benefits cannot be used to pay for fees of any type, such as delivery, service, or convenience fees" [108].

While the roll out of the pilot program was limited to certain states and designated retailers, the pandemic challenges of social distancing measures for safety further highlighted the importance of this allowance for online purchases. Transportation constraints often impacted people's access to grocery stores before the pandemic, and these challenges were multiplied by stay-at-home orders and social distancing measures. As of 5 January 2021, 47 states are currently participating in the SNAP Online Purchasing Pilot [108]. Post COVID-19, the expansion of this program will help SNAP recipients to access food in a timelier manner, especially for regions that have limited access to food retail outlets or limited mobility. The ARP will increase access to online purchasing for SNAP participants as well as invest in technology to modernize electronic benefit transfer (EBT) and support retailers in the program [109].

In particular, low-income older adults will benefit greatly from an expanded adoption of the EBT acceptance for online delivery of groceries. This older population often has barriers to mobility which impacts their access to fresh, healthy food. In a study conducted by the U.S. Department of Agriculture in 2020, older adult participants had "difficulty following their doctor's advice because they could not afford or obtain fresh produce and healthy proteins" [110]. Older adults are also less likely to receive food assistance due to barriers such as "low awareness that they are eligible, inability to navigate the application process, low benefit levels, and a sense of shame about the need to rely on assistance programs" [110]. A focus on increased visibility for eligibility, assistance for applying for benefits as well as expansion of delivery will increase the acceptance of the program among older adults to help mitigate future health risks.

*6.4. There Will Be Increased Innovations in Child Nutrition Programs*

Early in the pandemic, with the forced closure of in-person learning in schools across the United States, the importance of the federally funded Child Nutrition Programs became even more apparent (school lunch and breakfast, afterschool meals and snacks, summer food, WIC, and child care food). The National School Lunch Program (NSLP) is the second largest food and nutrition assistance program in the United States. In some households, reliance on the NSLP is a critical component of the family finances, as well as nutritional needs. School meals are a nutrition safety net for low-income children. According to a study, almost half (47%) of the children's daily energy intake was provided by the National School Breakfast and School Lunch Program meals [111]. In fiscal year 2019, the NSLP provided low-cost or free lunches to 29 million children per day during the school year [108].

When schools suspended in-person learning in March 2020, school districts across the country tackled the enormous challenge of continuing the school meals program through innovative ways, at a significant cost to their food service budgets. Waivers and exemptions to school meal regulations were enacted to provide flexibility in determining where and how school meals could be served during closures [112]. From creating congregate central community meal delivery sites, to delivering meals along school bus routes, schools spent

increasing amounts of money from their budgets for added packaging, PPE for employees, and more single serving items to provide needed meals for their students.

In addition, during the pandemic economic downturn, more families qualified for free or reduce priced meals due to changes in their financial status. As a result, school food budgets have been significantly impacted and many school districts will feel the negative financial strain for several years. Innovative solutions such as expanding meal service to seven days per week, offering grab-and-go meals in outdoor locations, and providing up to a week of meals at once have helped during the current crisis [112].

However, school nutrition programs will not be able to fill the meal gap among school-aged children during out-of-school time; P-EBT systems are an essential policy mechanism in the future [112]. In fact, with the extension of the P-EBT through the summer of 2021, "Congress is essentially running a pilot program" of extending school meals into a large summer feeding program, which "Biden officials have signaled an interest in making it permanent" [113].

The increased support for the National School Breakfast and Lunch programs—as well as Summer Meal programs—through recent legislation demonstrates a stronger interest in a universal school meal policy. Pre-pandemic, the USDA only permitted some schools that have a high poverty threshold among the student body to serve meals to all children at no cost, under the Community Eligibility Provision (CEP) [114]. As more research is conducted regarding the cost effectiveness of meals at no cost to children, the support for a universal school meal policy grows, but will need significant bipartisan support.

### 6.5. Innovations Will Be Adopted to Promote Food Security for Older Adults

During the pandemic, social distancing and restricting nursing home visits helped to protect the elderly from contracting the virus, but also negatively impacted food-insecure and low-income older Americans with an increased risk for malnutrition, hunger, and depression. Food and nutrition for seniors has a significant influence on how they live their lives and spend their days. In addition to the importance of the nutritional components, the socialization that seniors receive when dining with others is critical to their health and wellbeing. Many organizations such as senior centers and food banks pivoted to offering a delivery model, which can be expanded after the pandemic to reach mobility impaired individuals. However, maintaining some type of socialization is also important. Congregate meal settings are still an integral part of senior living, especially in communal neighborhoods or properties. Social isolation can increase mortality rates among seniors [87]; so communal dining is an important part of well-being and longevity. An important step in maintaining quality of life for seniors requires redesigning dining spaces to be more flexible and allow for more physical distancing once more of the population has become vaccinated and social distancing guidelines changes to allow for in person dining.

Expanding and continuing meal delivery options, with healthy fresh and frozen meals, in these communities will be crucial. Enhancing the in-unit dining experience, to mirror more of the room service model from hotels, will help to feed seniors if required to isolate; nonetheless, maintaining socialization is still important. Offering opportunities to livestream a cooking demonstration from an exhibition kitchen [115], or create virtual dining rooms to engage with other members of their community are options, but require the investment of broadband access and digital training skills. Creating a delivery model that allows for personal interaction with seniors, whether it is a quick chat in the doorway or a game or activity visible from the hallway, will help alleviate some of the isolation.

### 6.6. Mutual Aid and Community Solutions Will Rise in the Absence of Governmental Aid

Throughout 2020, increased awareness and community activism around mutual aid networks arose in response to "organizers trying to make sure their neighbors had access to what they need[ed] to survive given the failures of government to provide an intact social safety net" [116]. The practice of mutual aid has a long history in the United States; for example, the origins of the NSLP can be tied to survival programs initiated by the Black

Panthers to help underserved Black communities. The movement has been brought to the forefront during the pandemic and the social unrest regarding racism and systemic oppression in America.

Growing national awareness of inequalities, racial disparities, and increasing economic inequality over the course of 2020 helped foster community organizing to address the needs of the people. "Mutual aid networks often grow from the cracks formed incohesive public and private sector responses that fail to meet the needs of all people" [116]. Post COVID-19, the social inequity that helped strengthen this engagement will continue, making it likely that community activism through mutual aid networks will remain in place to help strengthen community wellbeing. "The mutual aid framework focuses on keeping social justice at the center while empowering those most directly affected" [117].

## 7. Conclusions

This article contributes to the ongoing dialogue concerning public health and safety in the post-COVID-19 landscape. The authors have brought together a plethora of sources, reviewing multidisciplinary literature across the fields of public policy, hospitality, labor studies, and health policy and management. Further, we enrich these summaries with insights from CIHF industry practitioners. Finally, we provide an encouraging outlook on how the pandemic could have long-term impacts, transforming our society for the better.

Throughout this work, we do not rely on quantitative analysis or primary research data. Rather, we provide a careful summary of published research and news articles thoughtfully selected by academic and industry experts affiliated with CIHF. Thus, while we did our best to provide a balanced and objective summary of this particular moment, the outlook of this article remains inherently subjective.

What has been clear, though, is that the disruptions to most of our lives have been significant. While we are facing challenges in the workplace, in particular due to a lack of child care and paid leave options for parents, we will also see benefits as workplaces become safer and better prepared. Telework will increase flexibility and reduce our carbon footprint. Policies and social norms will elevate the contributions of essential workers and may result in improved work conditions. The crisis has shed a light on racial and economic inequalities, which might lead to progressive changes.

Although the pandemic exacerbated food insecurity in the United States and in countries around the world, it also forced federal and local governments to increase flexibility and expand social safety net programs to help those in need. While many regulations will return to pre-pandemic status, there is hope for lasting, positive impacts. For example, a broad societal understanding of the need for equal access to healthy food has widened. As the conversation continues to unfold, there is optimism that some of these issues might become central to policymakers, whether through the extension of social safety net programs or by renewed focus on childcare tax credits to help lift families out of poverty.

Where do we go from here to best support healthy living post-COVID? We recognize there is both a tremendous need and a simultaneous opportunity to have policymakers, academia, and industry collaborate. Establishing policy, research and professional objectives is central to tackle the critical need for ongoing advances in the fields of labor and healthcare as well as healthy living more generally. It will ultimately inspire innovators—within organizations or as entrepreneurs—to continually progress, in policy and action, in the creation of an equitable, healthy future.

**Author Contributions:** Conceptualization, H.K., M.M.S. and N.R.Z.; methodology, H.K., M.M.S. and N.R.Z.; writing—original draft preparation, H.K., M.M.S., N.R.Z., E.V.-M.; writing—review and editing, H.K., M.M.S., N.R.Z. and E.V.-M. All authors have read and agreed to the published version of the manuscript.

**Funding:** This research received no external funding.

**Institutional Review Board Statement:** Not applicable.

**Informed Consent Statement:** Not applicable.

**Data Availability Statement:** Not applicable.

**Acknowledgments:** We would like to thank our advisory board members as well as the participants of the webinar "Healthy Futures: Transforming Our Lives Through COVID-19's Lessons" for very helpful comments and suggestions. We also thank Anna Wilkins for excellent research assistance. The research reported in this paper is not the result of a for-pay consulting relationship. Our employers do not have a financial interest in the topic of the paper that might constitute a conflict of interest. All remaining errors are our own.

**Conflicts of Interest:** The authors declare no conflict of interest.

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
