# Peer review of "How the COVID-19 Pandemic Will Change Workplaces, Healthcare Markets and Healthy Living: An Overview and Assessment"

_sustainability, doi:10.3390/su131810096_

Round 1
Reviewer 1 Report
The paper is informative and with excellent perspectives about the Covid 19 pandemic.
Author Response
We are thrilled to hear that the referee finds that we provide excellent perspectives about the covid-19 pandemic. Thank you very much.
Reviewer 2 Report
Dear Authors,
Review Comments on Manuscript entitled "How The COVID-19 Pandemic Will Change Workplaces, Healthcare Markets and Healthy Living: An Overview and Assessment" (sustainability-1294870):
Please consider the following points during the revision of the manuscript:
Your paper is very interesting and actual. However some improvements are needed, namely:
- In the Introduction section Authors should explain clearly the main aims of the work.
- The conclusion section needs to be more critical. Authors should explain, in a clear and complete way, what are the real (theoretical and practical) contributions of this paper to this field of research.
Author Response
Thank you very much for your comments. We are very happy to hear that you find the paper very interesting and timely.
We followed your suggestion and, using track changes, clearly explain now in the Introduction what the main aims of the paper are. Also please note that we submitted the work under the article category “Perspectives.”
Second, we also editing the Conclusion. Following your suggestion, we now discuss much more extensively what the contribution of this manuscript to the field of research are.

Reviewer 3 Report
Prior to my comments, I would like to thank you Sustainability that gave me the opportunity to review this perspective that has to do with COVID-19 and how will change our daily.
The COVID-19 pandemic has created unprecedented disruption for the global health and development community. Organizations fighting infectious disease, supporting health workers, delivering social services, and protecting livelihoods have moved to the very center of the world’s attention.
But they find their work complicated by challenges of access, safety, supply chain logistics, and financial stress like never before. Moreover, hundreds of millions of people have lived through lockdowns. Many have made the abrupt shift to working from home; millions have lost jobs. The future looks uncertain. We don't know when, or if, our societies might return to normal – or what kind of scars the pandemic will leave.
The entitled perspective on how society can adopt a healthier way of life based on the COVID-19. But what I believe is really missing from the paper is to highlight more the importance of IoT to the seniors in the post COVID world. Moreover, about telecommuting, I believe that authors should make a reference on the challenges of this type of work (factors that can interrupt teleworkers, how women affected by this type of work as they have to take care of their children, etc).
The abstract needs to be more analytical.
Author Response
Thank you very much for your comments. We agree with you and have made the following edits, using track changes in Word.
First of all, we edited the Abstract to make it more analytical.
Second, thank you very much for the hint about loT and senior living. We agree that smart nursing homes and senior living will play an increasingly important role in the future. Following up on your suggestion, we included an entirely new subsection in Section 5 “Senior Living in the Post-Covid World”, namely “5.2 The Potential for Internet-of-Things (IoT) Monitoring Solutions. “
Third, we also followed your suggestion and included a reference on the challenges of work from home, particularly with regard to women and difficulties related to child care possibilities in the United States.
